# ExerTrack—Towards Smart Surfaces to Track Exercises

**Biying Fu [1],\*,† , Lennart Jarms [1],† , Florian Kirchbuchner [1] and Arjan Kuijper [1,2]**

[1] Fraunhofer IGD, 64283 Darmstadt, Germany; lennart.jarms@igd.fraunhofer.de (L.J.);
florian.kirchbuchner@igd.fraunhofer.de (F.K.); arjan.kuijper@igd.fraunhofer.de (A.K.)

[2] Interactive Graphics Systems Group, Technische Universität Darmstadt, 64289 Darmstadt, Germany

\* Correspondence: biying.fu@igd.fraunhofer.de

† These authors contributed equally to this work.

**Abstract:** The concept of the quantified self has gained popularity in recent years with the hype of miniaturized gadgets to monitor vital fitness levels. Smartwatches or smartphone apps and other fitness trackers are overwhelming the market. Most aerobic exercises such as walking, running, or cycling can be accurately recognized using wearable devices. However whole-body exercises such as push-ups, bridges, and sit-ups are performed on the ground and thus cannot be precisely recognized by wearing only one accelerometer. Thus, a floor-based approach is preferred for recognizing whole-body activities. Computer vision techniques on image data also report high recognition accuracy; however, the presence of a camera tends to raise privacy issues in public areas. Therefore, we focus on combining the advantages of ubiquitous proximity-sensing with non-optical sensors to preserve privacy in public areas and maintain low computation cost with a sparse sensor implementation. Our solution is the ExerTrack, an off-the-shelf sports mat equipped with eight sparsely distributed capacitive proximity sensors to recognize eight whole-body fitness exercises with a user-independent recognition accuracy of 93.5 % and a user-dependent recognition accuracy of 95.1 % based on a test study with 9 participants each performing 2 full sessions. We adopt a template-based approach to count repetitions and reach a user-independent counting accuracy of 93.6 %. The final model can run on a Raspberry Pi 3 in real time. This work includes data-processing of our proposed system and model selection to improve the recognition accuracy and data augmentation technique to regularize the network.

**Keywords:** capacitive sensing; capacitive proximity-sensing; human activity recognition; exercise recognition; exercise counting; ubiquitous sensing; smart surfaces

## 1. Introduction

The concept of the quantified self has recently gained popularity, since miniaturized gadgets such as smartwatches or other fitness trackers are dominating the market. Integrated acceleration-based apps in smartwatches for step-counting and outdoor exercise monitoring are encouraging people to perform exercises. Although wearable devices were mostly used in the domain of human activity recognition, they still face lots of challenges due to the physical limitations of body-worn sensors. Maurer et al. [1] identified in their research that the exact location of sensor placement will strongly affect recognition performance and could lead to misclassification. Vision-based systems have gained much interest in the research field of human activity recognition, especially in the deep-learning era of recent years. The review by Zhang et al. [2] lists various applications of human activity recognition by leveraging vision-based methods. These remote installations such as camera installation can overcome the limitation of on-body sensor placement, but they may still raise privacy issues due to the visible input data stream. Other posed challenges are susceptibility to environment illumination change and occlusions. Therefore, we need a system to overcome these limitations and challenges. In this work,

we propose *ExerTrack*—a smart surface based on capacitive proximity-sensing to recognize eight fitness exercises including *push-up, sit-up, squat, segmental rotation, trunk rotation, swim, bridge*, and *quadruped*. A list of the targeted exercises is illustrated in Figure 1.

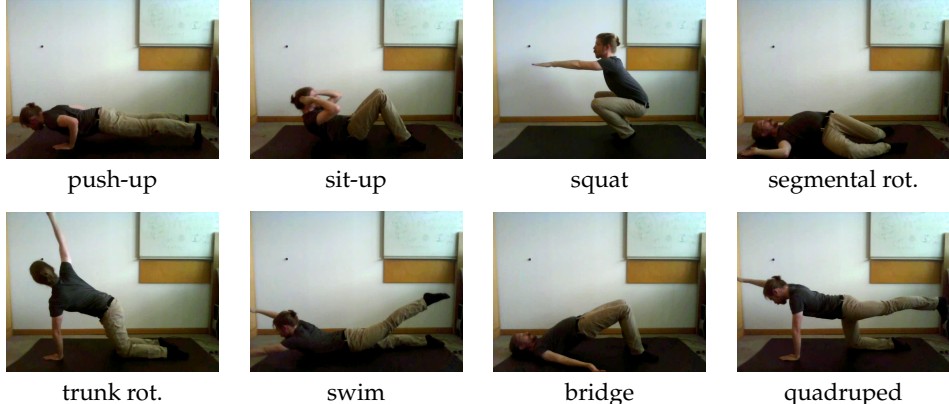

| push-up | sit-up | squat | segmental rot. |

| trunk rot. | swim | bridge | quadruped |

**Figure 1.** Eight exercises we recorded on our proposed sensing system.

The set of exercises is chosen such that we can show the advantage of our floor-based application compared to wearable devices. Further justification of choosing this set is to focus on whole-body exercises that can demonstrate the power of proximity-sensing over pressure-sensing. In the case of a *push-up* class, the chest movements to and away from the sensing surface are still visible in the signal without touching. Thus, the contributions of our work are as follows.

- We propose a floor-based sensing system using a capacitive proximity sensor to recognize eight strength-based sport activities.
- Due to the limited amount of labeled training samples, we demonstrated the ability of using data augmentation methods to regularize the classification network from over-fitting.
- We further demonstrate the advantages of using our proposed floor-based system to a single arm-worn accelerometer for the targeted set of whole-body activities.
- Owing to the simple network model architecture and the sparse resolution of our system, it can be run online on a Raspberry Pi 3.

The structure of our work is organized as follows: in Section 2 we introduce some relevant works in the field of exercise tracking. In Section 3 we describe the prototype of our proposed floor-based sensing system and clarify some design choices. Next, in section 4 we propose some data-processing methods regarding the behavior of the physical sensor. We further discuss traditional methods used to synthesize new training samples to increase the generalization ability of the classification network. A data-driven approach with hand-crafted features from time domains is used to train the inference model, such as a support vector machine (SVM), k-nearest neighbor (k-NN), or a Gaussian naive Bayes network (GNB). To reduce prior knowledge from human expert or domain knowledge, we also provide several architectures to train the network in an end-to-end learning fashion. In Section 4.4, we introduce a way to count the repetitions by using normalized cross-correlation with a template signal. In Section 5 we show the proof of concept by conducting a test study with 9 participants, each performing 2 full sessions and provide some first results achieved from the study. A comparison to a wearable device with acceleration data is drawn in Section 6. Finally, in Section 7 we discuss the limitations of our proposed system and provide some final thoughts to resolve part of these challenges and offer some future research directions.

## 2. Related Work

Encouraging people to regularly exercise is a well-researched topic in ubiquitous computing, especially using accelerometer sensor-based approaches with wearable devices [3,4]. Tracking and

recognizing the respective activities has successfully been implemented for various aerobic endurance training exercises, such as *walking, running*, or *cycling*. On the other hand, there is a limited amount of research on the topic of recognizing more stationary exercises, such as strength-based training or stretching, without the use of wearable sensors. These are just as important for a healthy lifestyle, as they prevent injuries and are essential for rehabilitation [5]. Typically, these exercises are harder to track than *walking* or *running*, as they rely on coordinated movement of specific body parts.

Chang et al. [6] used a glove equipped with accelerometer sensors to track free-weight exercises. They demonstrated that two initial measurement units (IMU), one attached to the right hand and one to the hip, can achieve recognition rates of up to 90 % for tracking nine different free-weight exercises. The classifiers used in their work consist of a naive Bayes classifier [7] and a hidden Markov model [8]. Morris et al. [9] introduced *RecoFit*, a system to automatically track repetitive, stationary exercises with a single arm-worn IMU. They conducted a user study with 114 participants resulting in over 146 exercise sessions. The proposed algorithm is grouped into segmenting exercise from intermittent non-exercise periods, recognizing the performed exercise and counting repetitions. They achieved a precision and recall rate greater than 95 % in identifying exercise periods and 96 % recognition rate of the correct exercises from up to 13 different exercises. For segmentation and classification, they rely on hand-crafted features that are fed into a support vector machine (SVM) classifier. Repetitions were conducted by peak counting on principle component analysis (PCA) [10] reduced accelerometer signal with the most variance. It is interesting to note, however, that the target activities only include exercises where the arm-worn IMU is always in motion.

Anton et al. [11] proposed a Kinect-based algorithm that provides real-time monitoring of physical rehabilitation exercises. They achieved 91.9 % accuracy in posture classification and 93.75 % accuracy in trajectory recognition related to how well certain activity is performed. For comparing the postures, a template-based matching score is used. Dynamic time-warping method (DTW) [12] is applied for trajectory comparison with pre-stored trajectories. Khurana et al. [13] proposed *GymCam*, a camera-based approach in an unconstrained environment, such as in a gym, to recognize, track, and count various workout exercises. Their proposed system can correctly segment exercises from other non-exercises with an accuracy of 84.6 % by tracking only repetitive movements commonly occurring during exercises. They can differentiate between exercise with a level of accuracy of 93.6 %, and count the number of repetitions within a deviation of ± 1.7 on average. Camera-based vision systems have made large progress in recent years with knowledge gained from the deep-learning domain. However, a camera system installed within the private or public sector still raises privacy concerns. In addition, computer vision methods on images or videos still require much computational power compared to other sensor data.

Wearables are quite precise in converting human activities to acceleration signals, since they are directly worn on the body. This can be a strong limiting factor in the same way, because it could hinder the natural movement of the body during exercises. Therefore, a remote system is more suitable to track the exercises. Sundholm et al. [14] proposed *Smartmat* to recognize physical exercises using a high-resolution textile pressure sensor matrix. They trained with 7 subjects for 10 exercises and achieved an overall accuracy of 88.7 % using a k-NN classifier. They also extended their approach to other applications, such as integration into a car seat for driver assistance [15]. Due to the high sensor resolution, the data-processing is performed offline and does not allow for real-time recognition and feedback on mobile consumer or other devices with limited computing resources. Similar commercial products based on piezoresistive pressure sensors that give individual feedback on yoga exercises have been developed in the project of SmartMat [16]. Therefore, we can observe a need for non-optical sensing systems to detect whole-body exercises.

In contrast to the high-resolution pressure mat, which requires close interaction with the body to input pressure, we propose a novel method of proximity-sensing with an active capacitive sensor mat. An active electric field emitted by the capacitive sensors can even measure body parts at a distance of 20 cm [17] from the sensing system, allowing it to distinguish more a distinctive pattern without direct

contact. For example, in the class *push-up*, we are able to detect the approaching chest movement, while a pressure sensor can only detect the pressure profiles from both hands that execute direct force on the mat. Capacitive proximity-sensing has found widespread use due to its flexibility and low cost [18], especially in the domain of human activity recognition (HAR). Among other things, capacitive sensors have been integrated into car seats for driver assistance [19], car seat occupancy detection [20] and into the floor for localization [21,22]. Haescher et al. [23] use their *CapWalk* system for the recognition of walking-based activities and achieve better results than accelerometer-based approaches, especially at lower walking speeds. Wearable solutions such as capacitive textile garments have been used to measure muscular activity while performing gym exercises [24]. Zhou et al. [25] extended this approach to monitor leg muscle activity.

As stated, research in the area of smart fitness mats is limited to pressure-sensor-based systems and to the best of our knowledge, no comparable system using capacitive proximity-sensing has been investigated for exercise recognition that entails whole-body movement detection. Thus, we are trying to fill this gap with our proposed mat-based system using capacitive proximity-sensing and to provide some interesting future research directions. We demonstrate the advantage of our proposed system with respect to a single arm-worn acceleration-based sensor and the ability to protect a user's privacy by processing the user data in an efficient way without uploading to any external server or device.

## 3. Hardware Setup

The chosen gym mat is a common consumer mat. Its dimensions are $190 \times 100 \times 1.5$ cm and it is made of synthetic, soft and form-preserving rubber. The OpenCapSense (OCS) toolkit [17] is used to develop the *ExerTrack* prototype. It is an open-source developing toolkit used to do rapid prototyping for capacitive sensing applications. The capacitive operation mode for our proposed application is called loading mode. In this operation mode, one side of the sensing electrode generates a low-frequency electric field. The distance of the single-sided electrode plate and the human body is measured through the change in capacitance coupling over the electrode to the ground. The operation principle is illustrated in Figure 2. If no object is present in the electric field, the other plate is thought to be at a distance of infinity. If an object is present, such as a foot is inside the electric field, the capacitive coupling will affect the charging and discharging characteristic on the sensing part. This affects the loaded capacitance. The coupling capacitance is measured by averaging the measurement for a small time window. Averaging sensor values over time reduces the noise coupled to the input signal. To further reduce the environmental coupling from the ground surface, the capacitive sensor applied in this work can use an active shield on the ground side. The function of the active shield is to load both the sensing plane and the opposite shield plane with the same electric potential. In this way, there is no direct coupling from the sensor electrode to the ground. This should reduce the parasitic coupling from the ground to the sensor. Thus, by applying the active shield, the different ground surface should not affect the sensor measurement. These positive effects of active shielding are depicted in Figure 3.

The operation frequency of our proposed sensing mat is 20 Hz. According to literature research, we identified that human body movements are contained within frequency components below 20 Hz and 98 % of the fast Fourier transformation (FFT) coefficient is contained below 10 Hz [26]. Thus, this operation frequency is sufficient to operate for the intended tasks of the whole-body sport activity recognition.

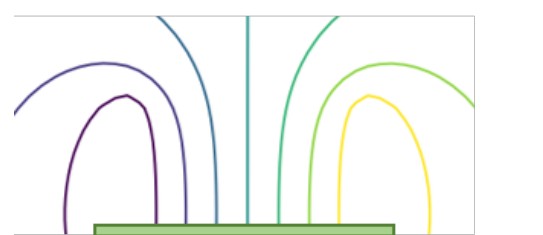
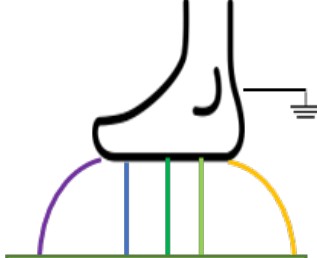

**Figure 2.** Approximate electric field lines in the case of loading mode capacitive sensing. If no object is present, the opposite parallel plate can be imagined to be in the infinity. When a human object is above the plate capacitor, the electric field lines will end on the foot as the opposite plate capacitor. Electric field lines are always orthogonal to the surface.

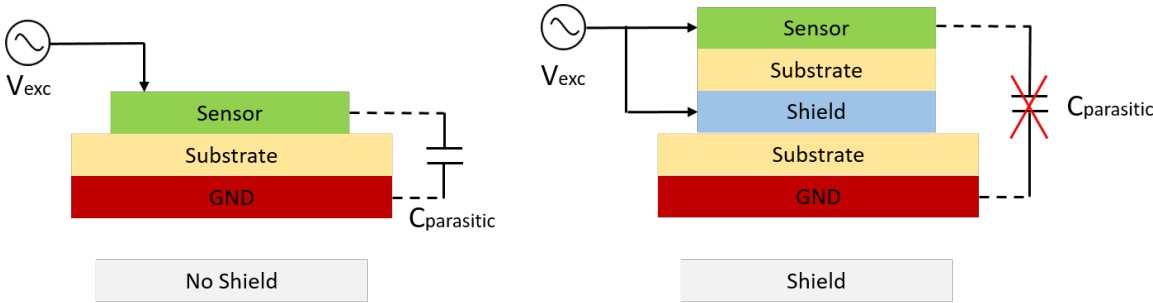

**Figure 3.** The working principle of an active shield to avoid the parasitic coupling from the environment.

Different electrode materials, including copper wire, conductive textiles, copper stripes and copper plates, were tested with respect to their sensitivity, signal strength and dynamic range. We decided to use shielded solid copper plates, since this allowed detection of floating body parts up to 15 cm above the mat. Details are provided in Table 1. Based on the sensing range, robustness, and signal quality, we observe the importance of an active shield. It aims at mitigating parasitic and environmental inferences and thus increases the signal-to-noise ratio for the true measurement. Therefore, the choice of using a double-sided copper plate is justified.

**Table 1.** Comparison of electrode materials. The option of shield is of vital importance to increase the sensing range and robustness. Therefore, the choice of use in this work is the double-sided copper plate with sensing and shielding side.

| Property | Double-Sided Plate | Sheet | Foil | Cable |
|---|---|---|---|---|
| Shielded | yes | no | no | no |
| Range | 15 cm | 5 cm | 3 cm | 0 cm |
| Robustness | rigid | soft | deformable | solid |
| Signal quality | excellent | noisy | noisy | good |
| Scalability | perfect | good | bad | good |
| Longevity | robust | short | short | robust |

The size of the double-sided copper plate as electrode was chosen with respect to the coverage of the sensing area and the detection range of distant object. The larger the electrode surface, the larger the detection range. To test different electrode positions, Velcro tape was sewn on a blanket and glued on the shielded side of the plate electrodes. This allows testing of different sensor placements while ensuring that the electrodes do not change position after continued use of the mat. As such, the mat itself remains portable and the blanket with the electrodes can be attached to other gym mats.

This is mainly designed to allow for reliable testing and is not intended as a commercial prototype. The prototype of our design can be seen in Figure 4.

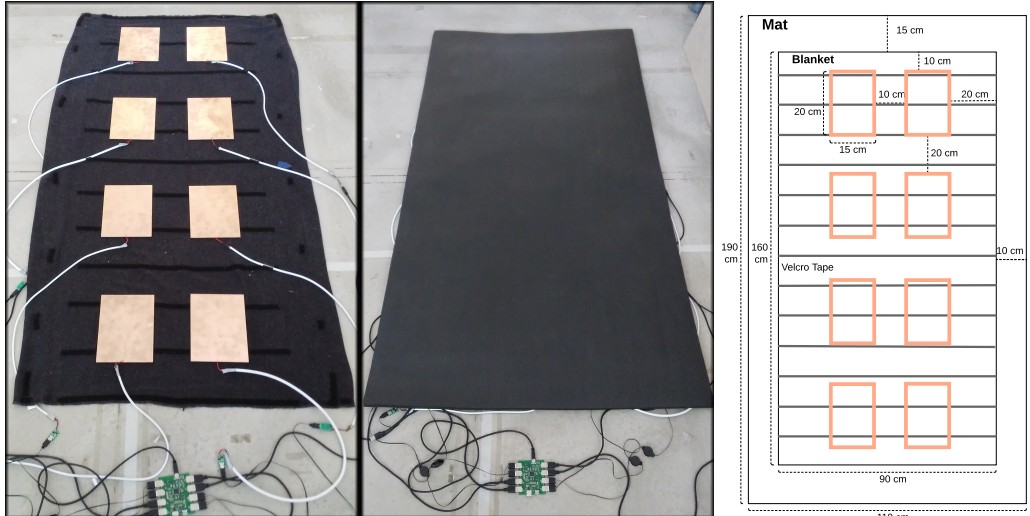

**Figure 4.** Prototype of our proposed sensing mat for sport exercise recognition by using eight capacitive proximity sensors. The symmetrical setup is used to allow the option for easy data augmentation process.

Regarding the sensor placement, initially the idea was to target different body parts with the sensors. However, to reduce the constraints on the user's orientation and posture, we decide to use the symmetrical layout depicted in Figure 4. This symmetrical layout can be easily leveraged for creating synthetic data, also known as data augmentation. This setup allows us to simply rotate and mirror the recorded data. This approach is considered to be useful while training end-to-end networks to reduce the over-fitting problem. The spacing between the electrodes is chosen to adapt more variation in height for different users.

## 4. Software Implementation

The system operates at 20 Hz, which corresponds to a time resolution of 50 ms. In Figure 5 such an input sample containing a repetitive movement is illustrated. The two visualizations are of the same signal. Line graphs are more intuitive to interpret when analyzing time series data. The image representation will be used by the convolutional neural network, which commonly operates on image data. To the right in Figure 5 the sensor identifiers are mapped to their location on the sports mat.

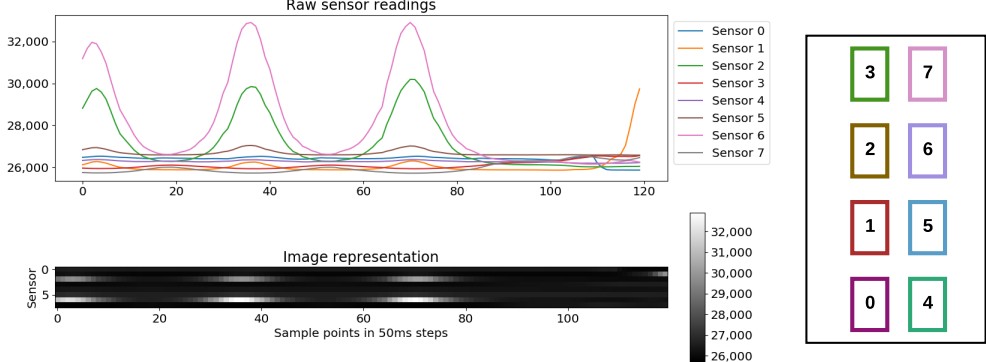

**Figure 5.** A raw sensory input of 6 seconds length and its corresponding sensor values. The image representation is just another representation of the same information as time series. The sensor numbering is depicted on the right.

The sensor numbering is not arbitrary, since it serves to fit the kernel filter type of the convolutional neural network architecture. Different filter types such as locally connected or dilated filter would modulate different sensor correlations. Data appears to have a clear structure when performing repetitive movements in succession but is subject to high variance depending on the location on the mat and exact placement of the body parts. In capacitive proximity measure, we measure the modulation of the electric field caused by capacitive coupling between the object interacting with the static electric field generated by a capacitive sensor. This electric signal will be sampled by the analogue-to-digital converter (ADC). The sensor value is defined by the static electric field generated by the sensor $x_0$ and the current environment coupling term $n(t)$ over the time, defined by Equation $y(t) = x_0 + n(t)$. The measurements of the OCS board itself are stable and there is only minimal noise present compared to a high signal-to-noise ratio (SNR). The true signal is thus a combination of body interaction and environment coupling $y(t) = x_0 + n(t) + b(t)$, where the term $b(t)$ stands for the interaction of the human object.

### 4.1. Data Preprocessing and Cleaning

In this work, the data-driven approach is selected over model-based approach. The reason is that there exists a very strong inter-person variation due to the performance of the exercises and the location of execution on the sensing mat. Therefore, it is not feasible to learn a generalized classifier with a fixed model. The performance of the classification models based on a data-driven approach is highly related to the underlying training data and its distribution. A clean data set with well-labeled data and a broad distribution to cover all the variability can improve the performance and the generalization ability of the trained classification model.

The capacitive measurement has a baseline which is dependent on the environment it is placed in. The baseline could be different in different locations. This physical limitation of capacitive measurement made our data-processing stage quite difficult. Therefore, we moved the data-normalization stage inside the first layer of the neural network architecture to overcome the problem of varying baseline values. No other cleaning is needed, since the signal-to-noise range from the human body is very high, i.e., $b(t) >> n(t)$, so that true modulation of the electric signal is dominating over noise.

### 4.2. Data Augmentation

As we discovered earlier, there is a lot of intra- and inter-user variation in the collected data. We need to collect a lot of annotated data to reflect this distribution. However, the data acquisition process for activity recognition is tedious and expensive. Extensive manual labeling is required. Thus, only a limited number of samples are available to describe the true sample distribution. Therefore, data augmentation is introduced to generate authentic and realistic synthetic signals. Data augmentation is a method to add prior knowledge into the data by transforming the data with methods that preserve the known label information [27] and that model variations of the data we expect. As it is heavily used in deep-learning methods for computer vision such as by Krizhevsky et al. [28] to improve the generalization error of the model, it is not that easy to adopt it to time series.

We distinguish between "domain-specific" and "general" augmentation methods. The former is bound to the system setup of *ExerTrack* and consists of interchanging the order of the incoming data sensor readings to reflect the symmetrical sensor setup. The operations are very simple: The sensor values from one sensor are swapped with those of another, which is illustrated in Figure 6. If we interpret the incoming data as an image, they represent mirroring the data at the horizontal and vertical axes. These operations mimic rotations of the user on the mat to some extent. We can argue that flipping the data is enough to model the user's orientation on the mat, as users naturally orient themselves along the mat to perform exercises. By introducing these variations, we expect to encode independence of the user's orientation on the mat.

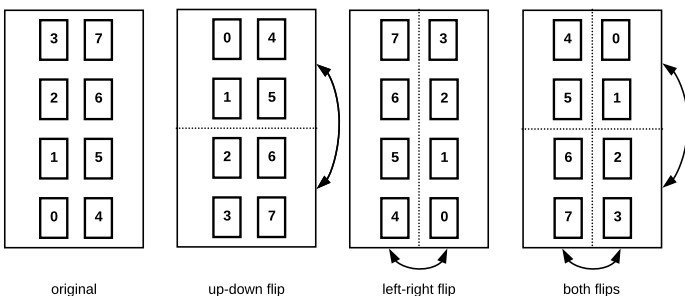

**Figure 6.** Flipping of the sensor data, which corresponds to rotating the user on the mat.

The general augmentation methods include common methods such as *jittering*, *scaling*, *magnitude-warping*, *cropping*, *permutating* and *time-warping* [29]. Due to specific sensor behaviors of our capacitive measurement, we can exclude jittering, permutating, and cropping. Therefore, the set of general modification just focus on magnitude and time-warping, to reflect the realistic sensor data. Magnitude-warping multiplies smoothly varying factors with the time series that can be modeled with piece-wise cubic polynomial functions around 1 as suggested by Um et al. [30]. We propose a similar approach to time-warping, by interpreting the polynomials around 1 as time intervals: if we accumulate them, we receive new indices which can be used to re-sample the original data and interpret them as samples from the original indices again. This results in smooth warps along the time axis. We can model coarse variations with lower-degree polynomials and fine variations with higher degree polynomials. This is illustrated in Figure 7, where a fine time warp is followed by a coarse time warp and subsequently a fine and coarse magnitude warp. The gray dashed line indicates no warp −−, the area above leads to stretching, while the area below squeezes the time series.

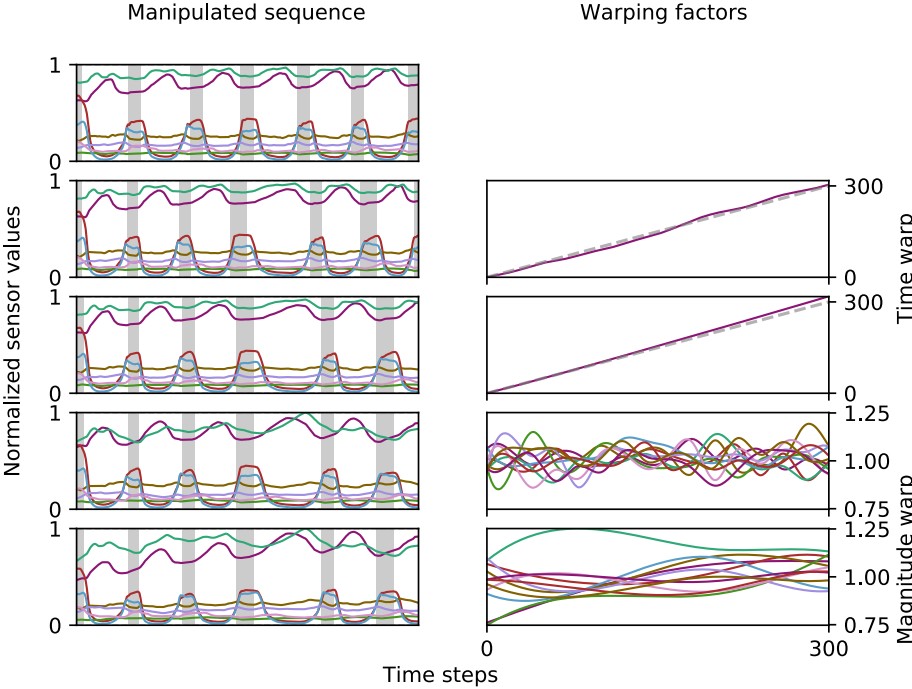

**Figure 7.** The process of applying multiple label preserving warps to the same sequence. The gray areas on the left denote breaks between exercise repetitions. For *time warp* a polynomial is accumulated and interpreted as time intervals to modify all sensor channels.

In our pipeline, the number of knots we choose for the piece-wise polynomials depends on the length of the time series. The transformations are applied to entire sets of exercises, which ensures

continuity of the time series. This also means that we treat the created variations as synthetic data. When evaluating models later, we only use this artificial data for training and not for testing.

In Table 2 the data augmentation methods we evaluated for different classifiers are listed. $\sigma$ controls the variance of the polynomials around 1, i.e., higher values lead to higher factors and consequently greater distortions.

**Table 2.** The tested data augmentation methods for which the evaluation will be carried out. $r \in \{0,\dots,3\}$ denotes the original orientation and the three flips. $M_f, M_c, T_f, T_c$ denote fine and coarse magnitude and time warps.

| Augmentation Method | Applied Transformations | Data Increase |
|:---:|:---:|:---:|
| Domain | $X_r$ with $r \in \{0,\dots,3\}$ | $1:4$ |
| General | $M_f(M_c(X, \sigma = 0.2), \sigma = 0.1)$ | $1:4$ |
| | $T_f(T_c(M_f(M_c(X, \sigma = 0.1), \sigma = 0.2), \sigma = 0.2), \sigma = 0.2)$ | |
| | $T_f(T_c(M_f(M_c(X, \sigma = 0.3), \sigma = 0.6), \sigma = 0.6), \sigma = 0.6)$ | |
| All | $X_r$ with $r \in \{0,\dots,3\}$ | $1:8$ |
| | $T_f(T_c(M_f(M_c(X_r, \sigma = 0.1), \sigma = 0.2), \sigma = 0.2), \sigma = 0.2)$ | |
| All Extended | $X_r$ with $r \in \{0,\dots,3\}$ | $1:16$ |
| | $M_f(M_c(X_r, \sigma = 0.2), \sigma = 0.1)$ | |
| | $T_f(T_c(M_f(M_c(X_r, \sigma = 0.1), \sigma = 0.2), \sigma = 0.2), \sigma = 0.2)$ | |
| | $T_f(T_c(M_f(M_c(X_r, \sigma = 0.3), \sigma = 0.6), \sigma = 0.6), \sigma = 0.6)$ | |

The "domain"-augmentation method only consists of the previously discussed data flips and as such has no transformations of the sensor values. $X_r$ with $r \in \{0,\dots,3\}$ denote the original orientation, left–right, up–down and both flips, respectively. As a comparison, the "general" method only includes concatenated magnitude and time warps with an identical increase in data size.

The other methods combine the domain and general approaches, applying time and magnitude warps to each of the flips. The process of the "all" method is visualized for one orientation in Figure 7 and is repeated for each of the flips. It can be viewed as the more conservative approach, as it only applies one transformation to each of the flips. "All extended" concatenates the domain and general methods and as such leads to the highest data-size increase.

The generated synthetic data is only used for extending the training data. Whenever we evaluate the predictive capabilities of the later described models we only evaluate against the original data, to make sure that the models do not simply learn the augmentations themselves. This is important, since negative learning is possible where the augmentations may *decrease* the performance of models [31], in cases they distort the data too much.

### 4.3. Processing Pipeline

Activity recognition is based on window segmentation of time series. The classification for each time window can be divided into three categories—distance-based, feature-based, and end-to-end training. Distance-based methods suffer from noise in the input data. Since there is high variation in the data even for the same user, depending on the exact location on the mat and the user's orientation, performing distance-based classification on the raw sensor signal did not seem to be promising. Another drawback for distance-based methods is the high evaluation time. As *ExerTrack* is meant to include an online application that either runs on a smartphone or a Raspberry Pi, this aspect is critical. If we, for example, examine dynamic time-warping (DTW), the evaluation time for a univariate time series of length $n$ is $O(n^2)$. This would have to be repeated for every sensor, which would drastically impair the reaction time of our final system. As such, we focused on a feature-based approach that includes hand-crafted and statistically relevant selected features in combination with conventional

classifiers like support vector machine (SVM), k-nearest neighbors (k-NN) and Gaussian naive Bayes (GNB) and as another alternative, automatic deep feature extraction with CNN.

First, the time series data need to be segmented to appropriate segments with local labels. According to a pre-study, a time window of 6 seconds and an overlap of 50 % is selected to compensate for recognition accuracy and real-time response of the sensing system. For a 6 s time window, it includes 120 sample points from each sensor. Hand-crafted feature selection is a tedious and time-consuming process. For this purpose, Christ et al. [32] propose their Python library *tsfresh* (Time Series Feature Extraction on basis of Scalable Hypothesis tests). They streamline the computation of 794 time series features and parametrization. We use this framework to extract a sub-amount of features used to train our classifiers. The list of extracted features, their formal definition and parameters are given in Table A1 in the Appendix. All features are extracted for each of the eight sensor channels, resulting in 120 features in total for a given time series segment. Most of these features originated from the time domain, simply because their computation is fast. By applying the principle component analysis (PCA) method we further reduced the feature dimensions by 60% (from 120 to 74) while conserving 98% of the information.

The suitability of conventional classifiers is further compared against deep-learning approaches based on models using convolutional neural networks. The main difference is that the conventional models, such as GMM, k-NN and SVM, rely heavily on prior expert knowledge and thus on generating hand-crafted features, while convolutional neural networks (CNN) are known for their ability to automatically extract deep, meaningful features from raw input samples. The process of extracting features becomes part of the optimization process for the whole model.

We investigate several models and hybrid models based on convolutional neural network structures. The reason we chose the convolutional neural network structure instead of other sequence-modeling structures like Long Short-Term Memory (LSTM) networks is because of the special sensor placement and its ordering. Instead of interpreting the input sequence as a time sequence, we interpret it as a two-dimensional image, with special correlation between sensors over time. The literature shows that different types of convolution filters are perfectly suitable for extracting local correlations. The parameters of a kernel filter are commonly written as fs@$k_x \times k_y$ in the illustration of model architectures. The filters are conceptually illustrated in Figure 8. In our case, a convolution in the x-dimension represents a temporal convolution, i.e., finding patterns across time. In the y-dimension, the values of the different sensor channels are convolved, allowing us to extract inter-channel features. We make use of dilation to encode relations between left–right aligned sensors, which allows us to keep the sensor data in one dimension. If we encode the sensor values in a $2 \times 4$ fashion we would not need the dilation, but would introduce another dimension, increasing the complexity of managing the data.

Batch normalization is applied directly to the input. This step is crucial for our proposed application. While we could normalize the data beforehand, we argue that this process enables the network to deal with new data more robustly and does not depend on the global normalization of the training data as much. We experienced heavy performance fluctuations when the test data had higher sensor readings than observed in the training data due to different baseline values according to various environment coupling. Applying the input batch normalization instead of normalizing the input data beforehand alleviated this problem. The output layer matches the number of classes plus a *none* class. A final global averaging pooling (GAP) layer is then directly connected to the SoftMax output, given the probability distribution over different classes.

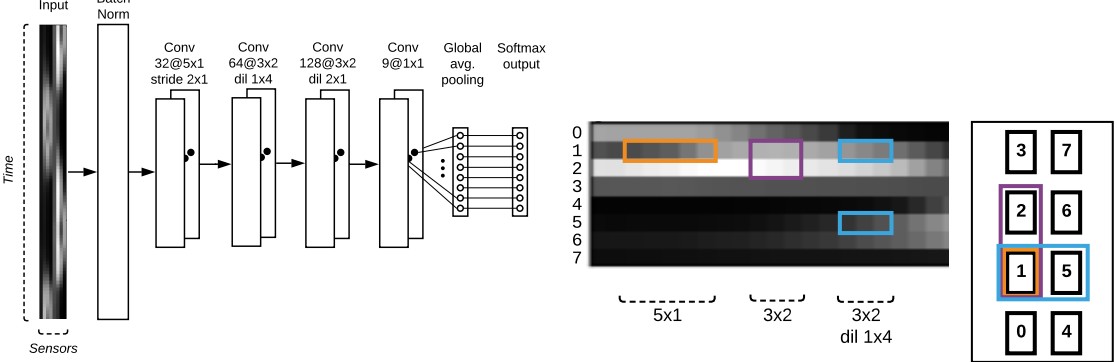

**Figure 8.** The three different kernel structures are used to extract different sensor correlations. Along the x-axis, features are extracted across time, and along the y-axis neighboring sensors are combined. These filters are the fundamental building blocks of a convolutional neural network. On the left, the sequential convolutional neural network (CNN) structure is shown with the classification layer on top.

In Figure 8 the sequential CNN architecture is depicted. After input normalization, a temporal convolutional is applied with a $5 \times 1$ kernel. The stride of $2 \times 1$ results in halving the input along the x-axis, to maintain the feature dimensions in the y-axis in the lower layers. As we want to give equal importance to features between neighboring electrode plates, we explore branching in our PBEF-CNN depicted in Figure 9 on the left. Compared to the sequential model, we introduce another convolutional block, but effectively do not increase the depth of the model, only the width. This architecture concatenates two different filter types to better catch the local correlations of close by sensors. Another tested model PBLF-CNN that implements late fusion as shown in Figure 9 on the right, where each branch consists of 3 convolutional blocks instead of one. How they get concatenated together investigates the way of concatenating more low-level features and fusing the decision layer at the very end.

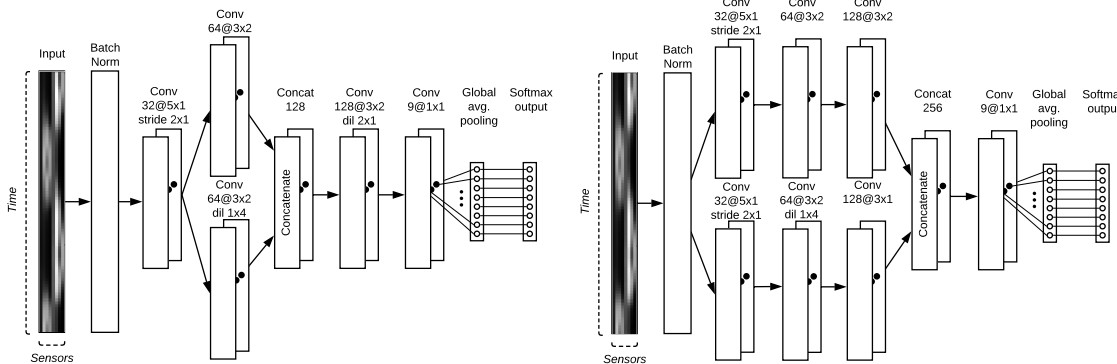

**Figure 9.** Parallel Branch Early fusion convolutional neural network (PBEF-CNN) model concatenating the structural information from spatially neighboring electrode is shown on the left side. The right architecture investigated the performance of the model by first maintaining the separated low-level feature extraction and concatenate the classification at the final stage. We name this structure the Parallel Branch Late Fusion convolutional neural network (PBLF-CNN).

To optimize the models, we use the Adam optimizer with a small learning rate of 0.0001. This is adjusted since we observed early that optimization was difficult and resulted in over-fitting to the training data very quickly. The models were implemented in Python with the *keras* [33] framework to easily create and test different architectures with its high-level API, using *tensorflow* [34] as backend. The models were tested and trained on a casual consumer PC with a *GeForce GTX 1060*.

*4.4. Repetition Counting*

The topic of finding exercise repetitions is commonly treated as another step after classifying. We extend the method proposed by Sundholm et al. [14] to fit our 8 channels sensor setup. The basic steps are described in the following order:

1. We first create templates for individual exercise from the training data.
2. We then move the template over the test data and find matches using a similarity measure.
3. We detect local maxima in the match score and count them as repetitions.

As discussed earlier in Section 4.2 regarding the data augmentation, we can flip the data, resulting in 4 possible configurations $f \in \{0, \ldots, 3\}$: original, left–right flip, up–down flip and both flips. This reflects all possible orientations a user can take above the sensing surface. To adapt Step (1) to deal with these variations of exercises, we must build a general template with re-oriented repetitions. While this alleviates the problem of averaging repetitions that vary greatly due to the orientation on the mat, it comes at the cost of having to flip and match the template three times to find repetitions later on.

With the calculated templates we can find matches by comparing them to an incoming signal using a similarity measure as described in Step (2). For this purpose, Sundholm et al. [14] used dynamic time-warping (DTW). In our experiments, this proved too computationally expensive for real-time execution and did not yield any better results than using cross-correlation. Hence, the zero-normalized cross-correlation (ZNCC) between a template and a signal of the same length is the better measure of similarity in our proposed system and can be calculated as follows:

$$f(X, T) = \frac{1}{N} \cdot \frac{1}{\sigma_X \cdot \sigma_T} \cdot \sum_{n}^{N} (X_n - \overline{X}) \star (Y_n - \overline{Y}) \tag{1}$$

where $n = (0, \ldots, 7)$ indicates the sensor channels. The benefit of using the zero-normalized variant is that the results will always lie within the interval $[-1, 1]$. We can use this to determine static thresholds to reject false peaks in step (3). The limitation of using the raw sensor data as input to our templates is that we need to know the orientation of the user and whether they are performing a left or right variation. We can simply flip the template in the remaining 3 orientations and calculate matches for each.

## 5. Experiments and Evaluation

The data is recorded in sets, i.e., all the exercises are executed consecutively. Transitions between exercises and breaks in between exercise repetitions are labeled as the *None* class. The length of sets ranges from 8 to 15 minutes, mostly depending on the length of breaks but also on the number of repetitions. For both the single and multi-user data set, the individual repetitions of each exercise are labeled manually during the recording. Furthermore, they are corrected afterwards with the help of a small interactive Python tool and the video recordings. This is necessary as mistakes were made while labeling every exercise repetition. The labeling tool was much more helpful for labeling individual repetitions, because the exact start and end times were clearly visible in the sensor signals. The video recordings were used to ensure that the start and end times of the exercise periods matched. The classes regarding the exercises are fairly balanced. The relative distribution is roughly the same for the single and multi-user data-sets. The single-user dataset is used to develop the user-dependent use-case, while the multi-user data are used to develop the user-independent use-case. The imbalanced data problem is handled in the model by introducing different weights for the classes depending on the class sample distribution.

The single-user data set consists of 12 sets performed by the same person. The sessions were manually labeled with a presenter stick during execution of the exercises. For leave-one-group-out cross-validation, we assign sets that were performed on the same day to the same group, as the exercises are performed very similar in these cases. This separation results in eight groups. We split

the data into six groups for training and two for testing. Hyperparamater optimization is performed by applying leave-one-group-out cross-validation on the training set.

The multi-user data set consists of 17 sets by 9 different participants. All users except one contributed two sets of data, which were recorded on different days for each user. The reason to not record the sets in succession is that we aim to capture as much intra-user variability in the data as possible, reflecting the findings from the previously conducted single-user study. The participants are asked to start performing the exercises in the randomly predetermined order. The reason for the random order is two-fold: We want to ensure that we capture more variety in the transitions from one exercise to another and that trained models do not specifically learn the order of exercises. One of the goals during the data collection is to not heavily influence the participants in their exercise form. Only when they clearly misunderstood the task or had further questions, more detailed instructions are given. For evaluation, we split the data into a training set consisting of 6 participants and a test set with the remaining 3 participants. For Hyperparameter estimation we perform leave-one-user-out cross-validation on the training set.

*5.1. Evaluation on Classification Performance*

Cross-validation is applied on the training data to optimize hyperparameters for these models. It is however also interesting to aggregate the performance when training on the different folds, as we can estimate the generalizability of the models. After this process, the model is fitted to the whole training set and evaluated against the test set. To ensure reproducible results, random seed for the training of the models and creation of augmented data were fixed. However, since the training of deep neural networks is computationally very complex and depends on non-linearities, it is not possible to reliably control the training. As such, the results are somewhat random. The three proposed architectures were trained in sequence, including the different proposed augmentation methods.

The box plots in Figure 10 denote the performance using leave-one-subject-out CV in the training set, which is also used for hyperparameter optimization. Since we do the latter, it could be that the optimized parameters are over-fitted to the training data and the models perform worse on the test set, which is why it makes sense to investigate. The performance on the completely unseen test set is indicated by the + scatter points. The best performance on the test set for the traditional models in the top row and the 2D-CNN-based models below is highlighted for comparison. We use the macro F1-score to compare the models. Since the class distribution is imbalanced with the *none* class dominating the other classes, the weighted F1-score is a better measure to compensate the ratio of recall and sensitivity in this multi-class scenario.

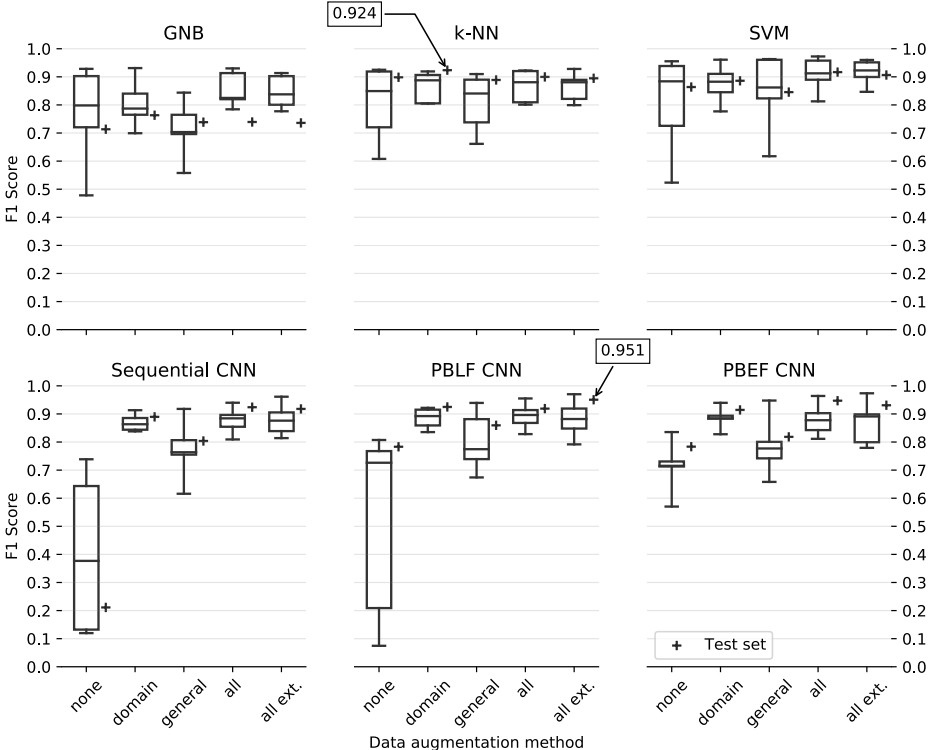

**Figure 10.** Box plot for the person dependent evaluation is shown on the left side, conventional classifiers in the upper row and CNN models below. The different data augmentation variants are on the x-axis for each model. The + indicates the performance on the test set.

For k-NN, the optimal parameter is found to be $k = 2$ with a distance-based weighting scheme, i.e., closer data points have a higher weighted vote for the assigned label. For the SVM the radial basis kernel(rbf) performed best, with an optimal penalty parameter $C = 11$ and $\gamma = 0.01$ ($\gamma$ defines the influence of single training examples).

One of the first trends we can observe is that introducing any method for data augmentation benefits the models, decreases the variance (indicated by the boxes) during leave-one-subject-out CV, and usually increases the weighted F1-score on the test set. In particular the domain method, which only flips the data and does not manipulate the measured values directly, increases the performance and decreases the variance.

The depicted confusion matrices in Figure A1 shows the correct predictions on the exercises. We compare the best conventional classifier with the best CNN model, based on their performance on the test sets, as highlighted in Figure 10. The k-NN appears to have the most trouble with the *None* class separation and with some exercises performed in the lying position. The PBLF model achieves better recognition rates for all exercises except for *quadruped*, which it confuses with *trunk rotation*.

The box plots for multi-user evaluation are illustrated in Figure 11 and the confusion matrices of the best performing models are shown in Figure A2. The conventional models perform a lot worse compared to the single-user evaluation, while the CNN architectures have only slightly worse performance. Thus, this confirms the presumption that hand-crafted features are often constrained and not robust enough to learn the inter-person variability. In the confusion matrix we can see that the SVM has trouble distinguishing exercises performed while lying, such as for (*bridge, segmental rotation* and *swim*), is similar to the k-NN in the single-user evaluation. The PBEF-CNN confuses barely any exercises with each other. It only performs slightly worse for the *push-up* exercise compared to the SVM and better for all others. As the separability from the *None* class is expected to be hard, it is acceptable to see confusion in this regard and we can conclude that the CNN outperforms the conventional classifiers by a great margin.

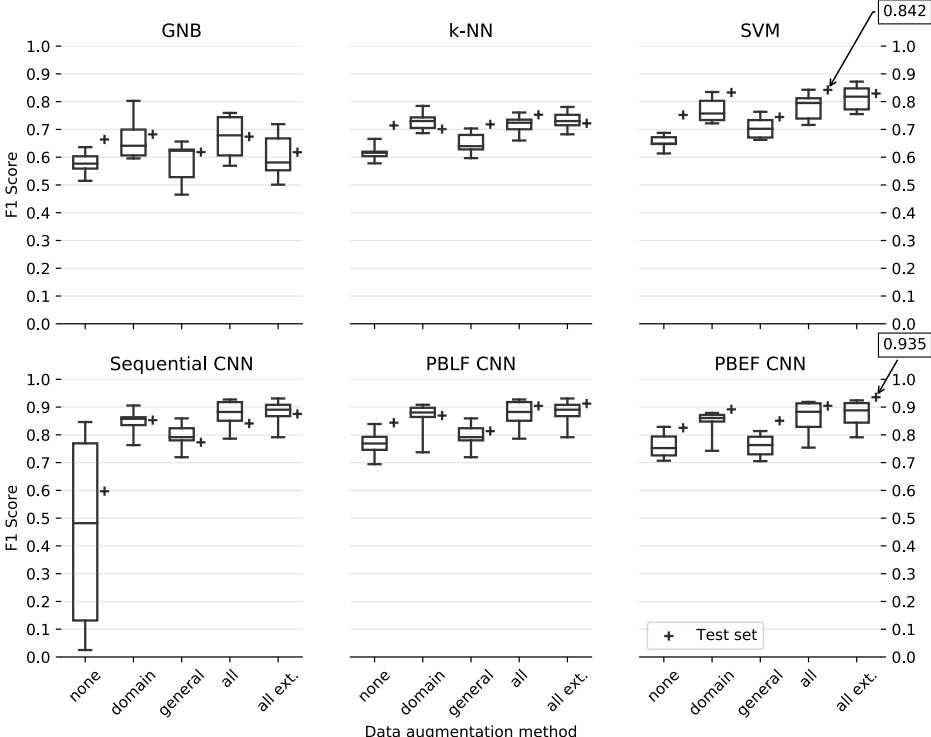

**Figure 11.** Box plot for the person-independent user evaluation is depicted on the right. The + indicates the performance of our test set.

For the proposed CNN models, it is not clear which performs best. Due to the random nature of training deep neural networks, minor performance differences can be expected when training the same architecture more than once. However, a general conclusion we can draw is that all CNN architectures profit from data augmentation, mostly so from the proposed domain method that only includes flips of the input data. With this method a reliably low variance on the training data is achieved with leave-one-subject-out cross-validation.

The test set F1-score of the CNN models is consistently higher than the average training CV F1-score, which is not always the case for the conventional classifiers. We can deduce that using more real data benefits the models the most, as often recommended to increase confidence in models [35]. Looking at the person-independent case, the sequential CNN tends to have worse than average performance on the test set, as such we would rather use one of the branching models.

*5.2. Evaluation on Repetition Counting*

For the single-user case, to evaluate the template matching, we perform leave-one-set/day-out CV on the 12 sets by the same participant resulting in 8 iterations, as we do not use a holdout set, i.e., we create templates on the training set and try to match repetitions in the unseen test folds. We can view repetition counting as binary decision making—if we detect a peak within the labeled segment that indicates a repetition, we count it as a true positive. False positive are peaks that are in repetition breaks or in the transition phases (before starting/after finishing an exercise set) where no repetition actually exists. We heuristically determined thresholds for the exercises to reject false peaks. Since we use normalized cross-correlation (NCC), a perfect match is 1. The thresholds are used to avoid false peaks which can occur in breaks between repetitions or in the transition from *None* to exercise and vice versa. A low threshold will lead to higher detection rates but might also introduce false positives, while a higher threshold will only detect exercises that are performed according to the template. The evaluation results can be seen from Table 3.

**Table 3.** Selected thresholds and calculated metric for repetition counting in the single-user case.

| Exercise | Threshold | Precision | Recall | F1 Measure |
|---|---|---|---|---|
| Push-Up | 0.6 | 1.0 | 1.0 | 1.0 |
| Sit-Up | 0.5 | 0.9884 | 1.0 | 0.9942 |
| Quadruped | 0.35 | 0.9839 | 0.9823 | 0.9831 |
| Bridge | 0.45 | 1.0 | 1.0 | 1.0 |
| Trunk Rot. | 0.45 | 0.9638 | 0.8859 | 0.9232 |
| Swim | 0.5 | 0.9841 | 0.9725 | 0.9783 |
| Squat | 0.45 | 0.9224 | 0.5879 | 0.7181 |
| Segmental Rot. | 0.5 | 0.9267 | 0.8932 | 0.9096 |
| Average | – | 0.9712 | 0.9152 | 0.9383 |

For the person-independent counting case, we compare the evaluation metrics used by Sundholm et al. [14]. For each person, we check which template from another person suits the best and count the repetition using the optimum template. Each positive count is for a peak in the normalized cross-correlation curve above certain threshold. However, we tried to find one general threshold for the same exercise throughout different participants to make the counting process more general and deterministic. In Figure 12 we calculated the repetition by using leave-one-participant-out approach. The true positive and true negative matches will be counted to calculate the evaluation metric like F1-measure as depicted on the y-axis.

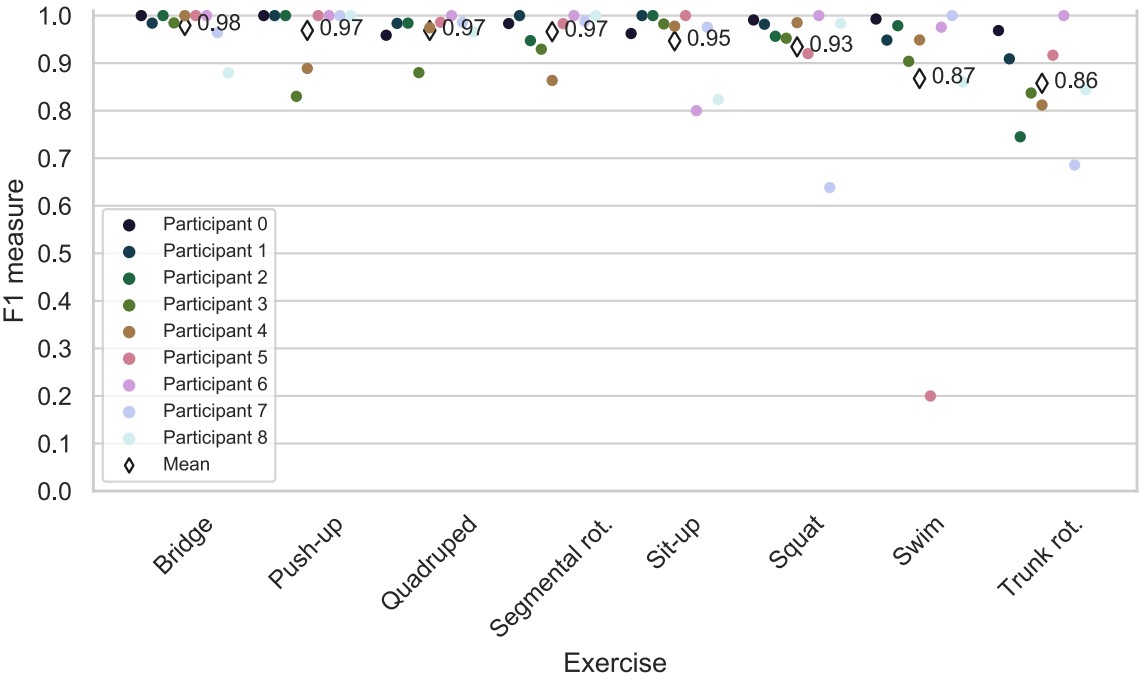

**Figure 12.** Exercise repetition counting results for each participant and exercise using template selection in the multi-user case.

We achieved a user-independent counting accuracy of 93.75 %. In case of exercises such as *bridge*, *push-up* and *quadruped*, we achieve even higher counting accuracy compared to Sundholm et al. [14]. One possible explanation can be the nature of our sensing principle. As for proximity-sensing, a clear signal of the distant chest movement can still be detected in mid-air even without direct contact to the mat, as opposed to pressure-sensing.

## 6. Comparison to Acceleration Data from Wearable

To demonstrate the advantage of our floor-based approach to single-worn acceleration data on the selected set of exercises, we simultaneously collected the sessions with users wearing a smartphone on the upper right arm. We make use of the 9-axis IMU data, including *linear acceleration, acceleration* and *gyroscope*. As the smartphone was attached to the upper right arm we assume difficulties when trying to disambiguate the exercises bridge and segmental rotation, where the user is lying on the floor and not actively moving the arms. All the other exercises include arm movements to some degree. This is also visible in the acceleration data depicted in Figure 13.

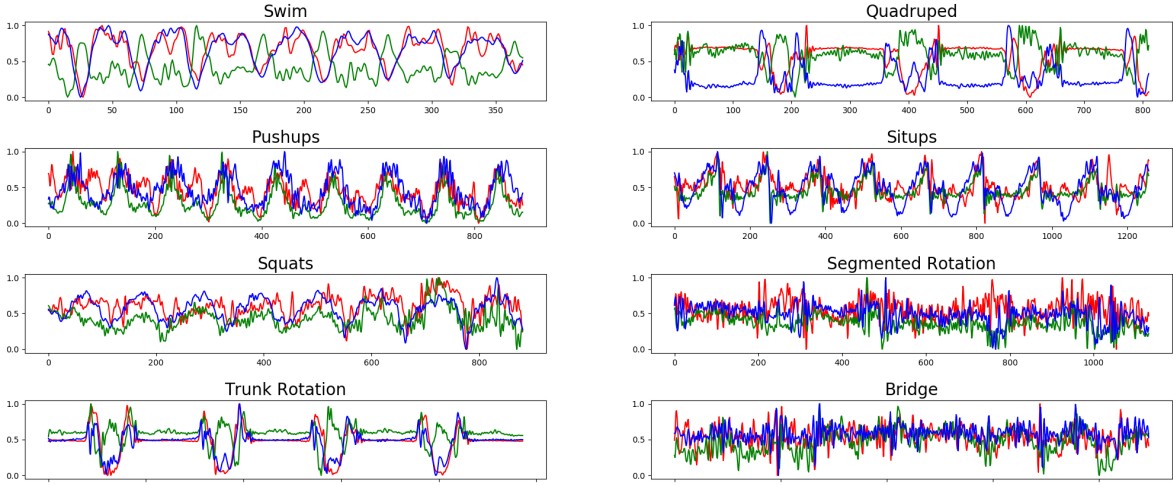

**Figure 13.** Acceleration data for the eight exercises from an entire sport session from one randomly selected participant. The red, blue and green curves represent the acceleration in x, y, and z directions, respectively. The x-axis is the sample time dimension and y-axis shows the normalized acceleration data. The class segmented rotation and bridge have the least modulation due to the dormant hand position.

A simple CNN with the same depth as the capacitive sensor CNN is implemented that only operates on the acceleration data as input. It similarly interprets the input as images, i.e., with dimensions of $120 \times 9$. The achieved weighted F1-score for the acceleration data is only 84.61 %. Compared to the capacitive-only solution, the weighted F1-score reduces to around 6.74 %. This outcome is expected, since the performance of a wearable is strongly correlated with placement and should be task-specific optimized.

## 7. Limitation and Discussion

In this work, we equipped an off-the-shelf sport mat with eight sparsely distributed capacitive proximity sensors to recognize eight workout activities. Despite the sparse sensor placement as opposed to other dense layouts, we trained several CNN structures to obtain good classification results. Regarding model selection, we have shown that CNN models can be applied in the case of only limited data that are available to reach a weighted F1-scores of 95.1% on a user-dependent study and 93.5% on a user-independent study. They outperform conventional classifiers based on hand-crafted features from the time and frequency domain: a k-NN reached 92.4% in the former study and a SVM 84.2% in the latter. Thus, we demonstrate the power of convolutional filters to catch locally correlated features across sensors.

As opposed to single wearable devices, the collection of eight capacitive sensors increases the performance of recognition, especially for classes in which the acceleration data does not provide enough information. However, compared to pressure-based sensing, the proximity-sensing further

enhanced the ability of providing signals from remote objects without touch. This suggests the possibility of detecting more fine-grained activities.

However, our proposed capacitive proximity-sensing system is by no means perfect. Regarding the hardware limitations, we need an active shield to reduce the environmental noise coupling to our sensing system from the ground surface. Further challenges lay in the nature of capacitive sensing, because the capacitive reading can be ambiguous. Further difficulties such as the problem of inter-person variability should be carefully addressed. This can be partly reduced by including a synthetic data-generation process. However, other ways of adaptation must be investigated to further improve the robustness and generalizability of the trained models.

Different baselines with respect to environmental noise also pose a problem to our proposed system. We target this challenge by using the normalization layer in the first stage of the neural network architecture, to make each input sample independent of the absolute values and leading the network to a faster convergence by restricting the input signal within a range of $[-1, 1]$. The problem of intra-class variation is partly targeted by the pooling layer in the convolutional neural network. By reducing the size in the time dimension, the network can modulate the slow and fast variability of users performing the same activity. This can replace the need for a time-consuming dynamic time-warping approach and thus allow us to build online applications.

## 8. Conclusions and Outlook

In this paper, we introduced *ExerTrack*, a smart sports mat for exercise recognition and tracking based on capacitive proximity-sensing. With only eight sparsely distributed electrodes we managed to achieve exercise recognition rates of 93.5 % for eight workout activities and a *None* class in a user-independent study and 95.1 % in a user-dependent study. These exercises are chosen such that it clearly demonstrates the advantages of our proposed system compared to wearable or other pressure-based sensing systems. Investigating different methods for creating synthetic data, we have shown that a system can heavily benefit if its sensor placement is chosen to allow for simple data augmentation methods (e.g., rotation). Applying the more general methods such as magnitude and time warp for time series manipulation and concatenating them has led to the best performance for the proposed convolutional neural network models on both single and multi-user data-sets. As the data acquisition process is costly and time-consuming, this is an important aspect to consider when evaluating prototypes, as it allows for more confidence in the evaluation of a system.

In the future, we will try to solve the above-mentioned challenges and limitations in Section 7 such as inter-person variability and intra-class variability. We are researching more advanced signal synthesizing approaches to reduce the differences in training and test data. Most machine-learning approaches have the assumption that training and test data originate from the same underlying distributions. This condition is very constrained and often does not hold for a test object with different physical or health states. Therefore, we must find a way to synthesize better data distribution, which partly covers the test distribution as well as reducing the difference between test and training data.

**Author Contributions:** Conceptualization and methodology: B.F., software, hardware: L.J., evaluation and data acquisition: L.J., writing—original draft preparation and final version: B.F., writing—review and editing: F.K., A.K. All authors have read and agreed to the published version of the manuscript.

**Funding:** This research received no external funding.

**Conflicts of Interest:** The authors declare no conflict of interest.

**Abbreviations**

The following abbreviations are used in this manuscript:

| | |
|---|---|
| CNN | convolutional neural network |
| GNB | Gaussian Naive Bayesian |
| HMM | Hidden Markov Model |
| k-NN | k-nearest neighbors |
| LSTM | Long short-term memory |
| PBEF | Parallel branch early fusion network |
| PBLF | Parallel branch late fusion network |
| PCA | Principle component analysis |
| SVM | support vector machine |

**Appendix A**

**Table A1.** The extracted features for every sensor channel, resulting in 120 features total for a time series segment.

| Feature | Parameters | Definition |
|---|---|---|
| Mean | – | $\overline{x} = \frac{1}{n}\sum x$ |
| Variance | – | $\sigma^2 = \frac{1}{n}\sum(x - \overline{x})^2$ |
| Skewness | – | $G_1 = \frac{n}{(n-1)(n-2)}\sum_{i=1}^{n}\left(\frac{x_i - \overline{x}}{s}\right)^3$ |
| Kurtosis | – | $G_2 = \frac{n}{(n-1)(n-2)}\sum_{i=1}^{n}\left(\frac{x_i - \overline{x}}{s}\right)^4$ |
| Sample Entropy | – | $-\log\frac{A}{B}$ |
| Absolute Sum of Changes | – | $\sum_{i=0}^{n-1}|x_{i+1} - x_i|$ |
| Autocorrelation | lag $l \in \{20, 40, 60\}$ | $R(l) = \frac{1}{(n-l)\sigma^2}\sum_{t}^{n-l}(X_t - \mu)(X_{t+l} - \mu)$ |
| Number of Peaks | $s \in \left\{\frac{20 \cdot \text{ws}}{4}, \frac{20 \cdot \text{ws}}{8}\right\}$ | – |
| FFT Coefficients | real, k $\in \{0\}$ | $A_k = \sum_{m=0}^{n-1} a_m \exp\left\{-2\pi i\frac{mk}{n}\right\}, k = 0,\dots, n-1$ |
| | absolute, k $\in \{0, 5, 15\}$ | |
| | angle, k $\in \{15\}$ | |

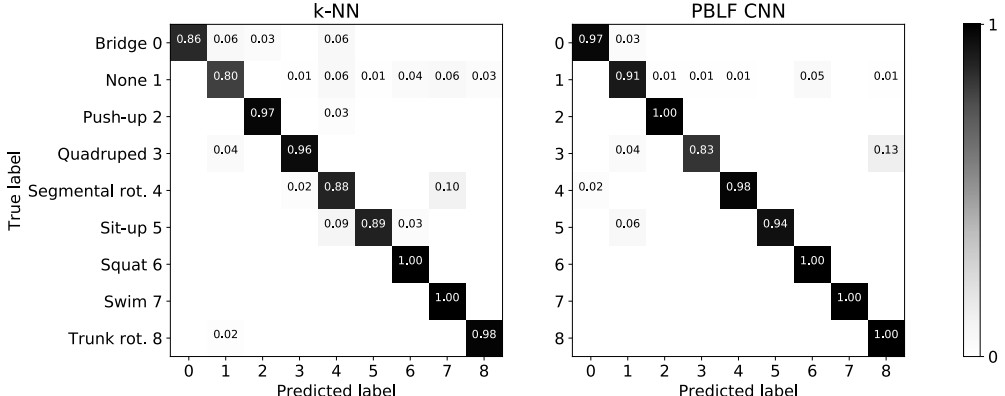

**Figure A1.** Confusion matrices of the models with the best performance on the test sets for the single-user evaluation is shown.

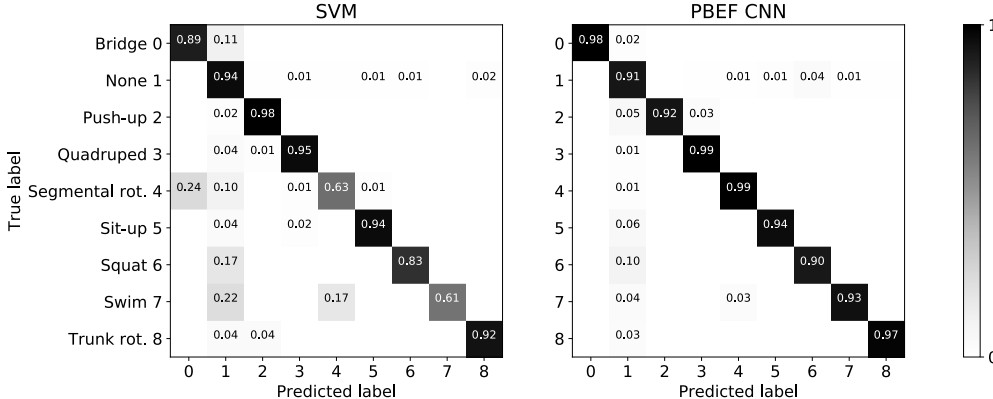

**Figure A2.** Confusion matrices of the models with the performance on the test sets for the multi-user evaluation is depicted here.

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
