# Peer review of "ExerTrack—Towards Smart Surfaces to Track Exercises"

_technologies, doi:10.3390/technologies8010017_

Round 1
Reviewer 1 Report
The authors should be commended for the submission of this technically sound scientific paper. The paper has some novelty components that highlight the strength of this paper. A stronger rationale and justification needs to be incorporated to ensure the gaps in the research are highlighted. This should be supported with a thorough grammatical check to ensure all sentences flow. The figures provided are clear.
The following are minor corrections to consider.
L10: change a to 'an off-the-shelf'.
L60: either say: as follows or in the following order
L63: change sentence to 'regarding the behaviour of the physical sensor'
L67: please improve sentence and get rid of the term 'introducing priors'.
L69-70: change sentence to 'there is a limited amount of research...'
L78-80: poor sentence - update / improve.
L80: Do you mean their proposed system can correctly segment exercises? please clarify? Perhaps you mean differentiate between exercise with a level of accuracy?
L88: write in full - does not NOT doesn't
L99: wide spread is one word widespread
L121: change the words effect to affect and effects to affects in L121.
L125: Update sentence to read as 'The function of the active sheild ....'
L146/ L239 /L301/L364: change 'don't' to do not
L148: change 'isn't' to is not
L158: Shorten sentence - Do you really need to say all this. The system operates at 20 Hz should suffice.
L180-181: These sentences are unclear.
L245: Avoid abbreviations '(there's) instead say there is a high....
L279: ..change 'of' to 'for' in this part of the sentence '...perfectly suitable for extracting local correlations'
Figure 8: Delete the word 'enable'
L285: change abbreviation to 'would not'
L289: change abbreviation to 'does not'
Author Response
Dear Reviewer 1,
I really appreciate the time and effort that you have dedicated to providing your valuable feedback on my manuscript. I am very grateful to your insightful comments and suggestions. I have been able to incorporate all changes to reflect the suggestions provided by you.
The basic wordings you proposed are all corrected. I rewrote part of the sentences to improve the meaning of unclear structures before, such as L67, L78-80, L180-181.
I reorganized the related work part and included several more related works to improve the flow of reading. A small conclusion was added at the end of related work part. In addition, I added two papers regarding acceleration-based wearable sensors and two more papers with respect to Kinect sensing.
Best Regards,
Biying Fu
Reviewer 2 Report
The topic covered by the manuscript fits well the scope of the journal and it is of potential interest for readers.
The state of the art should be improved by referring additional research works (not mandatory). The paper is well organized; the introduction and conclusion are supportive. All the other section provides sufficient information.
Author Response
Dear Reviewer 2,
I really appreciate the time and effort that you have dedicated to providing your valuable feedback on my manuscript. I am very grateful to your insightful comments and suggestions. I have been able to incorporate all changes to reflect the suggestions provided by you.
I reorganized the related work part and included several more related works to improve the flow of reading. A small conclusion was added at the end of related work part. In addition, I added two papers regarding acceleration-based wearable sensors and two more papers with respect to Kinect sensing.
Best Regards,
Biying Fu
Reviewer 3 Report
The paper presented an useful approach for tracking excercises. The method and the system were described in details, and the results are promising. Suggest to accept.
Author Response
Dear Reviewer 3,
I really appreciate the time and effort that you have dedicated to providing your valuable feedback on my manuscript. I am very grateful to your insightful comments and suggestions.
I reorganized the related work part and included several more related works to improve the flow of reading. A small conclusion was added at the end of related work part. In addition, I added two papers regarding acceleration-based wearable sensors and two more papers with respect to Kinect sensing.
Best Regards,
Biying Fu